

# Impact of chemical treatments on the molecular and stable carbon isotopic composition of sporomorphs

Yannick F. Bats[1], Klaas G.J. Nierop[1], Alice Stuart-Lee[1], Joost Frieling[1§], Linda van Roij[1], Gert-Jan Reichart[1,2], Appy Sluijs[1]

[1]Department of Earth Sciences, Faculty of Geosciences, Utrecht University, Princetonlaan 8, 3584 CB Utrecht, The Netherlands
[2]NIOZ Royal Netherlands Institute for Sea Research, 1797 SZ 't Horntje, the Netherlands
§ now at: Department of Earth Sciences, University of Oxford, South Parks Road, OX1 3AN, Oxford, United Kingdom

*Correspondence to*: Yannick F. Bats (y.f.bats@uu.nl)

**Abstract.** The stable carbon isotope composition ($\delta^{13}C$) of plants and algae is influenced by environmental factors, such as $p$CO2, water availability, and altitude. To effectively use the $\delta^{13}C$ of fossil material as proxies for these parameters, it is essential to understand the chemical and isotopic effects of diagenesis and conventional chemical treatments. In this study, we

subject various species of pollen and spores of higher plants to different chemical treatments simulating diagenesis and chemical alteration in the environment as well as palynological processing. We analyze changes in molecular and isotopic composition, using pyrolysis gas chromatography – mass spectrometry (MS), and both elemental analyzer and laser ablation isotope ratio MS, respectively. We find that saponification removes extractable and ester-bound lipids, which increases the $\delta^{13}C$ value of the sporomorphs. Treatment with HF and HCl removed most hydrolysable polysaccharides and proteins, causing

a drop in $\delta^{13}C$ values. Acetolysis produced aromatic-rich residual sporomorphs with the lowest $\delta^{13}C$ values compared to other treatments, likely representing the diagenetically resistant sporopollenin polymer. These findings imply a successional depletion of $^{13}C$ during fossil maturation, where aliphatic lipids are diagenetically removed in the process, until only the relatively $^{13}C$ depleted sporopollenin remains. To adequately compare fossil and extant sporomorph $\delta^{13}C$ values, we advise the use of HF-HCl and a lipid removal step other than acetolysis as palynological treatment, as acetolysis treats the material

non-uniformly. Lastly, laser ablation-IRMS shows promise for targeted isotopic analysis of individual specimens of various types of palynomorphs.

## 1 Introduction

The stable carbon isotopic composition ($\delta^{13}C$) of (sub-)fossil organic microfossils – such as pollen, dinoflagellate cysts, larval eggshells, acritarchs, and chironomid head capsules – are increasingly used as proxies in environmental and ecological

reconstructions, for example to infer $C_4$ versus $C_3$ grass abundance, atmospheric $p$CO$_2$, and diet (e.g., Kaufman & Xiao, 2003; Nelson et al., 2007; Schilder et al., 2018; Sluijs et al., 2018; van Hardenbroek et al., 2018). The mechanistic and statistical underpinning of $\delta^{13}C$-based proxies is often based on controlled growth experiments or field studies linking the isotopic





composition of the source organism to a specific variable (e.g., Loader and Hemming, 2004; van Hardenbroek et al., 2010; Hoins et al., 2015). However, when during taphonomy these organisms transform into their fossilizable remains, they undergo

*post-mortem* chemical alteration, also referred to as diagenetic alteration, which potentially affects their $\delta^{13}C$ composition (e.g., Benner et al., 1987; Loader and Hemming, 2004). In addition, these fossil remains typically undergo some type of palynological treatment before analysis, which often includes strong acids and might further affect their macromolecular composition and hence $\delta^{13}C$ values (e.g., Amundson et al., 1997; Müller et al., 2021). As a consequence, the molecular composition of a processed fossil is typically biased towards the most resistant organic compounds. Therefore, laboratory

experiments as well as modern calibration field studies must include similar chemical treatment to link the $\delta^{13}C$ values measured on extant and fossil sporomorphs.

Organic microfossils are comprised of organic macromolecular structures such as pollen and spore sporopollenin (often termed exine) (De Leeuw et al., 2006; Quilichini et al., 2015) that are resistant to most decay, except strong oxidation. However, in addition to the exine, extant sporomorphs often also contain pectin and (hemi)cellulose as well as attached lipids,

proteins and sugars (Fan et al., 2019). Such components all have their own inherent stable isotopic signature which may differ strongly in $\delta^{13}C$ from the mean exine value. Although this implies that their relative contribution potentially strongly impacts the overall stable isotopic composition, these more labile components are typically consumed or oxidized in the environment before sporomorphs are buried in sediments so that only the exine is preserved. Furthermore, it has been suggested that lipids may bind to the microfossils *post-mortem* in the sedimentary record (Van Bergen et al., 1993; De Leeuw et al., 2006; Versteegh

et al., 2007), influencing the integrated stable carbon isotopic signal.

Palynological treatment is commonly required to concentrate organic microfossils from sediments. Treatments typically include HCl to dissolve inorganic carbon phases and HF to dissolve siliceous materials. Acetolysis (acetic acid used together with acetic anhydride and $H_2SO_4$), KOH and NaClO are sometimes used, notably in terrestrial settings, to remove (amorphous) organic matter components and stain pollen walls. Several studies have shown that acetolysis strongly changes sporomorph

$\delta^{13}C$ (Amundson et al., 1997; Descolas-Gros and Schölzel, 2007; Müller et al., 2021). Also other steps, including HF-HCl and oxidants, may remove labile organic components and therefore potentially affect the $\delta^{13}C$ of the remaining palynomorphs (Müller et al., 2021). Such steps therefore also essentially simulate the effects of natural ageing and environmental oxidation.

After early $\delta^{13}C$ analyses of spores (Murphey and Nier, 1941), it took decades before pollen were targeted for $\delta^{13}C$ analyses, notably with the aim to distinguish $C_3$ from $C_4$ carbon acquisition pathways using total combustion isotope ratio mass

spectrometry (IRMS) (Amundson et al., 1997). Such early work needed large numbers of specimens for precise and accurate analyses. More recent work has optimized cryotrapping and subsequent release of $CO_2$ following combustion in an Elemental Analyzer (EA) so that much less material is needed for analyses (Polissar et al., 2009). Other devices utilize a continuous (He) flow approach, thereby further minimizing blank contamination before $CO_2$ is introduced into the IRMS. These include the use of a moving wire to introduce pollen – stuck to the wire – into an oxidation furnace, after which $CO_2$ is carried on an He

flow to an IRMS (Sessions et al., 2005; Nelson et al., 2007). More recently, a continuous flow technique was developed that carries molecular fragments derived from microfossil laser-ablation (LA) on a He flow into a capillary to a micro combustion





oven in which they are oxidized, followed by gas chromatography (GC) and IRMS (Van Roij et al., 2017). The latter technique allows for single specimen $\delta^{13}C$ analyses – similar to the ion microprobe technique used by Kaufman and Xiao, (2003) – and has been applied to pollen (Van Roij et al., 2017), lacustrine *Daphnia* resting eggshells (Schilder et al., 2018) and marine

dinoflagellate cysts (Frieling et al., 2023; Sluijs et al., 2018).

Here, we aim to contribute to the existing literature regarding chemically induced $\delta^{13}C$ change of raw sporomorphs by carrying out a series of experiments on raw pollen derived from multiple species. We apply acetolysis, HF-HCl, and saponification to study the individual effects of these common palynological treatment steps and also evaluate this in relation to similar transformation that might occur during taphonomy (diagenesis). We assess the character of chemical alteration

resulting from the treatments using pyrolysis gas chromatography – mass spectrometry (GC-MS). Finally, we analyze sporomorphs using conventional EA-IRMS as well as LA – nano combustion (nC) – GC – IRMS to assess analytical consistency.

## 2 Materials and Methods

### 2.1 Samples

Six sporomorph species were selected for this study. Concentrated pollen samples of a $C_3$ grass (*Lolium perenne*), a $C_4$ grass (*Zea mays*), a gymnosperm tree (*Pinus echinata*), and two angiosperm trees (*Betula pendula*, *Eucalyptus globulus*) were obtained from Greer Labs (Lenoir, USA). A spore sample (*Lycopodium clavatum*) was obtained from Baldwins (London, UK). For the rest of this article the species are referred to by their genus name.

### 2.2 Treatments

Three different treatments were carried out on each sporomorph species in triplicate. These treatments included HF-HCl, a common acid treatment in palynology to concentrate sporomorphs from mineral samples, saponification to remove ester-bound fatty acids (cf. Nierop et al., 2019), and acetolysis to remove polysaccharides and proteins to yield sporopollenin only (Erdtman, 1960; Hesse and Waha, 1989).

*Saponification*

In accordance with Nierop et al. (2019), sporomorphs were saponified in 1M KOH in MeOH at 70 °C for 2 h in an oven. After cooling to room temperature, the solution was acidified to pH 3 using 2N HCl. Then dichloromethane (DCM) was added and the whole mixture was thoroughly shaken using a Wilten Vortex Genie apparatus. After phase separation, both the aqueous and the organic layers were removed, and both water and DCM were added to purify the remaining sporomorphs. After

thoroughly shaking by the vortex apparatus followed by phase separation, both layers were removed again and this procedure was repeated twice. MeOH was added to the sporomorphs to remove traces of both water and DCM. This was done twice after which the remaining sporomorphs were dried using a very gentle stream of $N_2$.





*HF-HCl*

Samples of 200 mg untreated pollen were treated with 15 mL 38–40% HF. The samples were placed on a platform shaker for 120 minutes and centrifuged for 5 minutes at 2200 rpm. The supernatant was removed with a glass pipette and the pellet disintegrated on a vortexer. 15 mL of 30% HCl was added to each, followed by a further centrifuge step and separation of the liquid fraction. Distilled water rinses and centrifugation steps were repeated until the samples tested pH neutral, after which they were dried in an oven at 60°C for 72 hours and subsequently stored at room temperature until analyses.


*Acetolysis*

Samples of 200 mg untreated pollen were treated with 5 mL 99.6% acetic acid, which were then centrifuged for 5 minutes at 2000 rpm. 5 mL of a mixture of pure acetic anhydride and 95-97% sulfuric acid (9:1 volume/volume) was added and the samples were placed in a hot water bath at 100°C for 5 minutes, and from there transferred to a cold water bath for 5 minutes.

Following centrifugation, the solution was decanted, and the vials vortexed. Four rinsing steps were carried out, adding 12 mL of water, centrifuging, decanting and vortexing. Samples were then dried in an oven at 60°C for 72 hours and stored at room temperature until analyses.

## 2.3 Pyrolysis-gas chromatography-mass spectrometry (Py-GC-MS)

Analytical pyrolyses were carried out on a Horizon Instruments Curie-Point pyrolyser. Samples (typically 0.5 mg) were pressed

onto Ni/Fe Curie point wires and subsequently heated for 5 s at 600 °C. The pyrolysis unit was directly connected to a Carlo Erba GC8060 gas chromatograph and the products were separated by a fused silica column (Varian, 30 m, 0.32 mm inner diameter) coated with CP-Sil5 (film thickness 0.40 µm). Helium was used as carrier gas. The GC column was directly connected to the pyrolysis unit through a splitless injector set at 280°C. The oven was initially kept at 40°C for 1 min, next it was heated at a rate of 7°C/min to 320°C and maintained at that temperature for 15 min. The column was coupled to a Fisons

MD800 mass spectrometer (mass range m/z 45–650, ionization energy 70 eV, cycle time 0.7 s). Identification of compounds was carried out based on their mass spectra using a NIST library or by interpretation of the spectra, by their retention times and/or by comparison with literature data.

All samples were run at least in duplicate. Semi-quantification of the compounds identified was performed by integration of peaks using one or two target fragment ions. Correction for this to match total ion chromatogram (TIC) peaks

was achieved using the relative response of the fragment ions relative to TIC peaks by integration of well resolved TIC peaks in chromatograms of the samples. The standard (milled oak roots) run was used to check the performance of the Py-GC-MS. The relative abundance of each pyrolysis product was calculated from the sum of all integrated and corrected peak areas. All correction factors and relative abundances can be found in the Supplementary Data. These data reflect averages of all replicates per pollen and treatment types.






### 2.4 Stable carbon isotopic analyses

#### 2.4.1 EA-IRMS

Of each untreated and treated replicate, 50 µg was weighed in tin capsules for bulk $\delta^{13}C$ analysis on an elemental analyzer (Thermoscientific Flash 2000) coupled to an IRMS (ThermoFinnigan Delta V Advantage IRMS; short: EA-IRMS) at NIOZ

Royal Netherlands Institute for Sea Research. Series of analyses were bracketed by blanks (empty capsules), and Vienna Pee Dee Belemnite (VPDB) calibrated internal standards (nicotinamide, $\delta^{13}C$ -33.48‰; and a sediment standard graphite quartzite (GQ), $\delta^{13}C$ -26.68‰) for calibration to the VPDB scale and to correct for drift. The standard deviation of the GQ standard (0.07‰) represents the analytical error of the EA-IRMS method and is incorporated into the total uncertainty of the sporomorph $\delta^{13}C$ measurements. Considering other authors (e.g., Hemsley et al., 1993) have suggested acetolysis to treat material non-

homogeneously, the acetolyzed *Pinus*, *Lolium*, and *Zea* replicates were measured in triplicate (i.e., a triplicate of the triplicates), to assess the treatment's uniformity.

#### 2.4.2 LA-nC-GC-IRMS

One sample for each species and treatment combination (i.e., one of the triplicates) was blindly chosen and analyzed using the approach first described in Van Roij et al. (2017): laser ablation, nano combustion gas chromatography and isotope ratio mass

spectrometry (LA – nC – GC – IRMS; for the rest of this article abbreviated to LA-IRMS). Sporomorph grains from each sample were pressed onto a nickel disc using a hydraulic press. A polyethylene (PE) standard (IAEA-CH-7, $\delta^{13}C$ -32.15‰) was added, and the disc placed into an ablation chamber specifically designed for the purpose (Van Roij et al., 2017). Individual grains were targeted using a deep ultraviolet (DUV) ArF laser (COMPex 102; Lambda Physik, Göttingen, Germany, and, COMPexPro 102; Elemental Scientific Lasers LLC, Bozeman, USA, see Supplement Table S6 for which laser was used per

sample) to fragment the organic matter, with the ablated particles then being combusted in a micro furnace and the released CO2 transported to a GC and IRMS (ThermoFinnigan Delta V Advantage IRMS). At least five PE standard measurements were done with each LA-IRMS run, to calibrate $\delta^{13}C$ values using standard bracketing. Van Roij et al. (2017) suggested mismatches between EA- and LA-IRMS measurements to possibly result from single standard bracketing, which prevents correcting for linear mass bias over the full range of $\delta^{13}C$ data. Therefore, to test the linear mass bias of the LA-IRMS method,

some samples (Supplement Fig. S8) were additionally corrected with a new VPDB calibrated internal polylactic acid (PLA) standard (Premium Eco Clear Bags, ClearBags, North Las Vegas, USA; -12.63‰). Furthermore, Van Roij et al. (2017) showed measurements to be reliable (i.e., small offset from the value determined by EA-IRMS analyses and smaller standard deviation) with peak areas > ~2 Vs. Therefore some measurements were done by ablating multiple grains.

The standard deviations of the LA-IRMS measurements represent both inter-specimen variation within the samples

– which is averaged out with the bulk EA-IRMS method – and the analytical error of the LA-IRMS method. The internal analytical error is predicted using the standard deviation of PE measurements with a similar peak area as the respective sporomorph sample (Van Roij et al., 2017) and will be compared with the spread in values observed for the samples.



## 2.5 Data analysis

All data analyses were conducted in R (R Core Team, 2023) using the *dplyr* (François et al., 2023), *tidyr* (Wickham and Girlich,

2024), *ggplot2* (Wickham, 2016), *ggridges* (Wilke, 2024) and *boot* (Canty and Ripley, 2022) packages. For the EA-IRMS $\delta^{13}C$ measurements, the total uncertainty for each species/treatment combination was determined by combining the standard deviation ($\sigma$) of the replicate measurements with the analytical error (0.07‰) using quadrature addition ($\sigma_{tot} = \sqrt{[\sigma_{analytical}^2 + \sigma_{replicates}^2]}$). Treated and untreated EA-IRMS samples were compared using student's *t*-test.

LA-IRMS $\delta^{13}C$ measurements of each species/treatment combination were weighted according to their peak area (Vs),

for two reasons. Firstly, the peak area relates directly to the amount of ablated material in a single measurement (Van Roij et al., 2017), which implies a weighted mean more accurately represents the true sample mean, and, secondly, measurements become more precise as peak areas increase (Van Roij et al., 2017; Frieling et al., 2023), which implies that measurements with a higher peak area are statistically more meaningful. The weighted $\delta^{13}C$ measurement of each species/treatment combination were bootstrapped (*n* = 1000), to elude effects of non-normality in further statistical testing. Bootstrapped means

and standard errors of the mean (SEM) were used to compare measurements statistically between the treated and untreated LA-IRMS samples and between LA- and EA-IRMS using Welch's *t*-test. For each species/treatment combination, note that the EA-IRMS data is based on triplicates that underwent the treatment separately, whereas the LA-IRMS data comes from a single, randomly chosen triplicate. Therefore, Welch's *t*-test assesses the probability that a randomly drawn LA-IRMS sample matches the EA-IRMS based population mean, which encompasses the range of $\delta^{13}C$ values across the three separate

treatments. Lastly, standard deviations between LA-IRMS sporomorph and PE measurements were compared using an F-test.

## 3 Results

### 3.1 Sporomorph pyrolysis products upon treatment

*Untreated*

The pyrolysis products of the sporomorphs studied consist of an array of molecules broadly divided into lipids, aromatic

compounds and polysaccharides/proteins-derived components; the latter two are here grouped because of their similar hydrolysable nature. The pyrolysis products of the sporomorphs consist predominantly (65% – 81%) of lipids, between 5% and 36% of aromatics, and between 1% and 23% of polysaccharides and proteins (Fig. 1 and 2, and Supplement Fig. S1).

For lipids, fatty acids such as $C_{16}$ and unsaturated $C_{18}$ fatty acids dominate all untreated sporomorphs (Fig. 2). Other lipids such as odd numbered *n*-alkenes and *n*-alkanes in the range of $C_{25}$-$C_{31}$ are observed in the pollen of *Betula*, *Zea*, and

particularly in those of *Lolium* (Supplement Fig. S2). *Pinus* pollen contain some even numbered *n*-alcohols ($C_{24}$-$C_{28}$) (Supplement Fig. S2 and Table S2).

Polysaccharide-derived products predominantly represent levoglucosan (Supplement Fig. S3 and Table S3). Anhydropentose and levomannosan represent pyrolysis markers of the monosaccharides xylose and mannose (Supplement





Fig. S3). The relative abundance of polysaccharide-derived pyrolysis products is smallest for intact *Lycopodium* spores (~1%)
and largest for *Zea* pollen (~22%), with similar intermediate values for the other pollen types. Nitrogen-bearing pyrolysis
products include benzyl nitrile, methylbenzyl nitrile, indole and 3-methylindole, and are detected in all species (Supplement
Fig. S3), although in small amounts (maximally ~2% for *Eucalyptus*).

For aromatics, typical sporopollenin-derived pyrolysis components are the propionic acid and vinyl (i.e., ethenyl)
containing compounds phloretic acid, hydrocaffeic acid, hydroferulic acid, 4-vinylcatechol, 4-vinylphenol and 4-vinylguaiacol
(Supplement Fig. S4 and Table S4). The latter two are identified for all species, whereas 4-vinylcatechol only for *Lycopodium*.
Phloretic acid is detected in *Pinus* pollen in small amounts, while being absent from all other sporomorph species. Hydrocaffeic
and hydroferulic acid are detected in relatively large amounts in *Lycopodium* spores, while absent from other species.
Furthermore, the carbonylbenzenes benzaldehyde and acetophenone are consistently observed for all species (Supplement Fig.
S4). Other aromatic components present in all species are the methoxyphenols guaiacol (2-methoxyphenol), 4-methyl- and 4-
ethylguaiacol (Supplement Fig. S4). The alkylbenzenes styrene, toluene and benzene are also detected for all species
(Supplement Fig. S4). Alkylphenols detected in all species consisted of phenol, 2-methylphenol, 3/4-methylphenol and 4-
ethylphenol (Supplement Fig. S4).

*Saponification*

Saponified sporomorphs lose most of their free and ester-bound lipids. Moreover, a large relative increase of polysaccharide-
derived components is seen upon saponification, to values of 11% for *Lycopodium* spores, up to 87% for *Lolium* and 89% for
*Zea* pollen (Fig. 2 and Supplement Fig. S1). Also the N-bearing components increase relatively (e.g., to ~8% for *Eucalyptus*).
A relative increase of aromatic compounds is also witnessed upon saponification, but to a smaller degree than is the case for
polysaccharides, suggesting some aromatic compounds are removed by saponification.

*HF-HCl*

Most, if not all polysaccharides are removed from the sporomorphs by HF-HCl treatment. Also the N-bearing components are
not detected anymore (Supplement Fig. S3 and Fig. 1 for *Lolium*). Lipids and aromatics remain similarly abundant in the
chemical make-up of the sporomorphs (Fig. 2).

*Acetolysis*

Acetolysis removes most, if not all *n*-alkanes, *n*-alkenes and alcohols, and reduces the amount of fatty acids substantially. Most
polysaccharides are removed upon acetolysis, and N-bearing compounds are also not detected anymore. Only *Zea* retains
relatively high amounts of polysaccharides after acetolysis (Fig. 2). A large relative increase of aromatics is observed upon
acetolysis. Especially the relative amount of alkylbenzenes and phenol compounds increases in respect to their untreated
counterparts (Supplement Fig. S4).





**Figure 1.** Pyrolysis-gas chromatograms of *Lolium* pollen that were: a) untreated, b) saponified, c) HF-HCl-treated and d)

acetolyzed. Colors reflect the retention time intervals with predominantly aromatic (red), polysaccharide-derived (blue) and lipid (green) pyrolysis products.



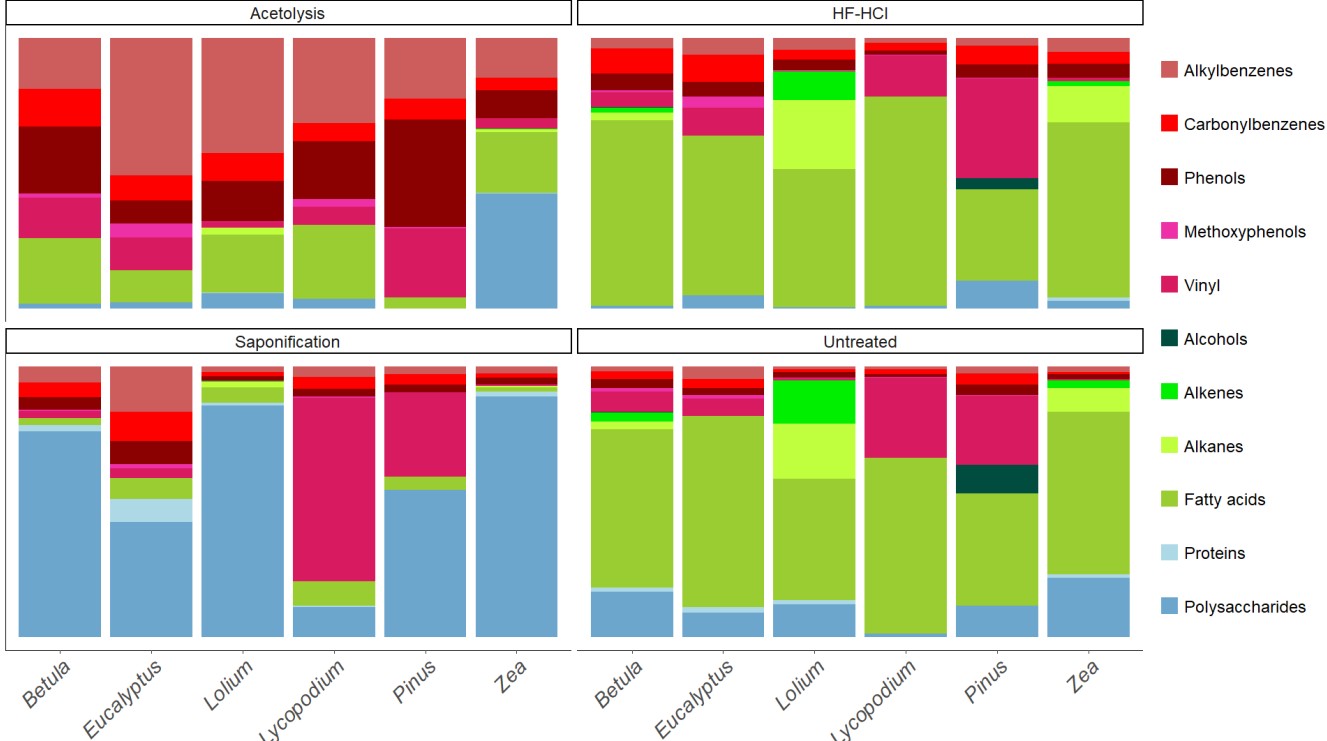

**Figure 2.** The composition of the various aromatic (alkyl- and carbonylbenzenes, methoxyphenols, phenols and vinyl containing compounds, red tints), polysaccharide- and protein-derived (blue tints), and lipid (*n*-alcohols, *n*-alkanes, *n*-alkenes and fatty acids, green tints) pyrolysis products for each species-treatment combination.

## 3.2 Sporomorph carbon isotope values

### 3.2.1 EA-IRMS

*Untreated*

The sporomorph mean $\delta^{13}$C values of the six studied species are presented in Table 1. All untreated sporomorph species have significantly (t-test, all $p < 0.05$) different mean $\delta^{13}$C values compared to each other, except for *Lolium* and *Eucalyptus* ($t = -1.52$, df = 4, $p > 0.05$) (Supplement Table S5, Fig. 3).

*Saponification*

Saponification significantly increases $\delta^{13}$C values relative to untreated samples for *Betula*, *Lolium*, *Eucalyptus* and *Zea* (*t*-test, all $p < 0.05$; Table 1). In contrast, the $\delta^{13}$C values for *Pinus*, and *Lycopodium* do not significantly differ between saponified and untreated pollen (t-test, all $p > 0.05$; Table 1) (Fig. 3).





*HF-HCl*

The HF-HCl treatment significantly (t-test, $p < 0.05$) decreases $\delta^{13}$C values across all pollen types compared to untreated
sporomorphs (Table 1, Fig. 3). It should be noted that for *Lycopodium* and *Pinus* only a single replicate was measured.

*Acetolysis*

Acetolysis significantly decreases $\delta^{13}$C values for all sporomorphs compared to the untreated sporomorphs, except for *Zea*
(Table 1, Fig. 3). The triplicates show a substantial range in $\delta^{13}$C values (Fig. 3), especially for *Zea* (-13.5 to 22.0‰), which
is the reason there is no statistically significant difference between the acetolyzed and untreated *Zea* sporomorphs.








**Table 1.** The EA-IRMS stable carbon isotope ($\delta^{13}$C) measurements of each species-treatment combination, and the respective $\delta^{13}$C difference with untreated samples of the same species. A student's *t*-test was performed to compare these differences. Only a single replicate underwent treatment for HF-HCl treated *Lycopodium* and *Pinus*, and therefore no *t*-test was performed for these samples. * = p<0.05, ** = p<0.01, *** = p<0.001.

| | EA-IRMS results | | | Difference versus untreated, *t*-test | | |
|---|---|---|---|---|---|---|
| **Species** | **Treatment** | **$\delta^{13}$C** | **SD** | **Difference (‰)** | ***t*-value** | ***p* value** |
| *Zea* | Untreated | -12.09 | 0.22 | | | |
| *Lycopodium* | Untreated | -29.78 | 0.08 | | | |
| *Pinus* | Untreated | -28.9 | 0.10 | | | |
| *Betula* | Untreated | -26.02 | 0.07 | | | |
| *Lolium* | Untreated | -28.04 | 0.13 | | | |
| *Eucalyptus* | Untreated | -27.85 | 0.09 | | | |
| *Zea* | HF-HCl | -14.91 | 0.31 | -2.82 | -7.44 | 0.002** |
| *Betula* | HF-HCl | -28.97 | 0.19 | -2.95 | -14.28 | 0.0001*** |
| *Lolium* | HF-HCl | -30.81 | 0.34 | -2.77 | -7.65 | 0.002** |
| *Eucalyptus* | HF-HCl | -29.56 | 0.08 | -1.71 | -13.93 | 0.0002*** |
| *Lycopodium* | HF-HCl | -29.73 | NA | NA | NA | NA |
| *Pinus* | HF-HCl | -30.07 | NA | NA | NA | NA |
| *Zea* | Saponification | -10.81 | 0.08 | 1.28 | 5.47 | 0.005** |
| *Lycopodium* | Saponification | -29.75 | 0.07 | 0.03 | 0.24 | 0.82 |
| *Pinus* | Saponification | -28.91 | 0.08 | -0.01 | -0.08 | 0.94 |
| *Betula* | Saponification | -25.77 | 0.09 | 0.25 | 2.25 | 0.09 |
| *Lolium* | Saponification | -27.33 | 0.11 | 0.71 | 4.19 | 0.01** |
| *Eucalyptus* | Saponification | -27.48 | 0.08 | 0.37 | 3.06 | 0.04* |
| *Zea* | Acetolysis | -18.03 | 3.9 | -5.94 | -1.52 | 0.2 |
| *Lycopodium* | Acetolysis | -31.95 | 0.23 | -2.17 | -8.89 | 0.0009*** |
| *Pinus* | Acetolysis | -32.50 | 0.24 | -3.60 | -13.86 | 0.0002*** |
| *Betula* | Acetolysis | -31.18 | 0.30 | -5.16 | -16.81 | 0.00007*** |
| *Lolium* | Acetolysis | -31.14 | 0.28 | -3.10 | -10.04 | 0.0006*** |
| *Eucalyptus* | Acetolysis | -31.90 | 0.26 | -4.05 | -14.68 | 0.0001*** |

### 3.2.2 LA-IRMS

Table 2 shows the results of the LA-IRMS $\delta^{13}$C measurements as compared to the EA-IRMS measurements. Table 3 shows the F-test results of the LA-IRMS standard deviations compared to the analytical error. LA-IRMS samples that are additionally corrected with PLA do not show significantly different results as compared to EA-IRMS, in respect to when they are solely



corrected using PE (Supplement Fig. S8), although there is a linear mass bias involved with the LA-IRMS method (Supplement Fig. S9).

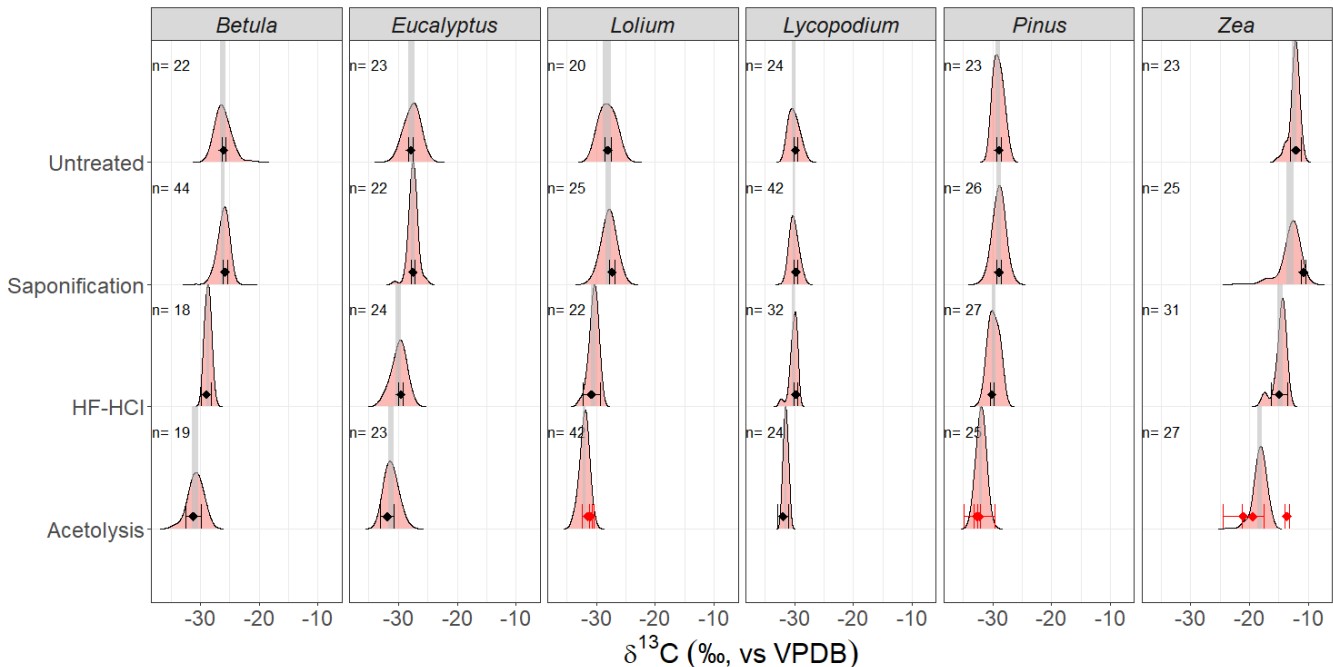

**Figure 3.** Mean stable carbon isotope ($\delta^{13}C$) and 95% confidence interval (CI) of the EA-IRMS measurements (black points and error bars), and $\delta^{13}C$ distributions of the LA-IRMS measurements (density plots), for each species-treatment combination. Grey shaded areas represent the 95% CI of the mean of the LA-IRMS measurements. Numbers above density plots represent the sample size (i.e., number of LA measurements) of each LA-IRMS sample. The red points and error bars represent the mean stable carbon isotope ($\delta^{13}C$) and 95% CI of triplicates of replicates, showing the non-homogeneous treatment of acetolysis. Note that for HF-HCl treated *Lycopodium* and *Pinus* only a single EA-IRMS measurement has been done (a single replicate), and therefore their respective 95% CIs are based on the 0.07‰ analytical error (see Methods).

*Untreated*

For each species, the LA-IRMS based $\delta^{13}C$ means of untreated specimens are not significantly different from the $\delta^{13}C$ means obtained using EA-IRMS (*t*-test, all $p > 0.05$; Table 2). Only for *Zea* and *Lolium* standard deviations are significantly larger than the analytical error (F-test, $p < 0.05$; Table 3).

*Saponification*





For *Zea,* the $\delta^{13}$C means obtained from saponified specimens using LA-IRMS are significantly (*t*-test, $p < 0.05$; Table 2) lower

than those obtained using EA-IRMS, with a mean difference of 2.27‰ (Fig. 3). The mean LA-IRMS $\delta^{13}$C values do not significantly differ between saponified and untreated sporomorphs (*t*-test, all $p > 0.05$; Table 2), even though the EA-IRMS method did reveal a difference for *Zea*, *Lolium* and *Eucalyptus* (Sect. 3.2.1.). Standard deviations are significantly larger than the analytical error for *Betula, Eucalyptus, Zea* and *Lolium* (F-test, $p < 0.05$; Table 3).

305        *HF-HCl*

For each species, the LA-IRMS based $\delta^{13}$C means of HF-HCl treated specimens are not significantly different from the $\delta^{13}$C means obtained using EA-IRMS (*t*-test, all $p > 0.05$; Table 2). Standard deviations are significantly larger than the analytical error for *Eucalyptus, Zea* and *Pinus* (F-test, $p < 0.05$; Table 3).

310        *Acetolysis*

For each species, the LA-IRMS based $\delta^{13}$C means of acetolyzed specimens are not significantly different from the $\delta^{13}$C means obtained using EA-IRMS (*t*-test, all $p > 0.05$; Table 2). All acetolyzed samples have a significantly lower $\delta^{13}$C than the untreated samples for LA-IRMS (Table 2). Standard deviations are significantly larger than the analytical error for *Betula, Eucalyptus, Zea* and *Lolium* (F-test, $p < 0.05$; Table 3).






**Table 2.** The LA-IRMS stable carbon isotope ($\delta^{13}$C) measurements of each species-treatment combination, the respective $\delta^{13}$C difference with untreated samples of the same species, and the difference with each sample's corresponding EA-IRMS measurement. Welch's *t*-tests were performed to compare these differences. * = p<0.05, ** = p<0.01, *** = p<0.001.

| LA-IRMS measurements | | | | | | Welch's *t*-test (between treated and untreated) | | | | Welch's *t*-test (between LA- and EA-IRMS) | | | |
|---|---|---|---|---|---|---|---|---|---|---|---|---|---|
| Species | Treatment | Mean $\delta^{13}$C (‰) | SD (‰) | n | SEM (‰) | Difference (‰) | *t* value | df (*t*-test) | *p* value | Difference (‰) | *t* value | df | *p* value (*t*-test) |
| *Betula* | Acetolysis | -30.8 | 1.1 | 19 | 0.33 | -4.7 | -10.9 | 42.68 | 6.1E-14*** | -0.40 | -0.91 | 2.74 | 4.4E-01 |
| *Eucalyptus* | Acetolysis | -31.2 | 1.0 | 23 | 0.24 | -3.5 | -8.6 | 30.59 | 1.3E-09*** | -0.71 | -2.00 | 2.51 | 1.6E-01 |
| *Lolium* | Acetolysis | -32.0 | 0.75 | 42 | 0.13 | -3.9 | -11.67 | 7.13 | 6.7E-06*** | 0.90 | 2.92 | 2.13 | 9.2E-02 |
| *Lycopodium* | Acetolysis | -31.5 | 0.4 | 24 | 0.08 | -1.4 | -7.2 | 17.06 | 1.3E-06*** | -0.46 | -1.89 | 2.18 | 1.9E-01 |
| *Pinus* | Acetolysis | -32.0 | 0.69 | 25 | 0.17 | -2.9 | -12.2 | 46.76 | 4.5E-16*** | -0.52 | -1.73 | 2.36 | 2.1E-01 |
| *Zea* | Acetolysis | -18.2 | 1.06 | 27 | 0.23 | -5.9 | -20.3 | 60.92 | 7.6E-29*** | 0.2 | 0.05 | 2.03 | 9.6E-01 |
| *Betula* | HF-HCl | -28.7 | 0.51 | 18 | 0.13 | -2.3 | -7.7 | 15.91 | 1.0E-06*** | -0.27 | -1.18 | 2.43 | 3.4E-01 |
| *Eucalyptus* | HF-HCl | -29.9 | 1.2 | 24 | 0.26 | -2.3 | -5.38 | 32.80 | 6.2E-06*** | 0.38 | 1.42 | 3.84 | 2.3E-01 |
| *Lolium* | HF-HCl | -30.5 | 0.78 | 22 | 0.19 | -2.3 | -6.437 | 21.00 | 2.2E-06*** | -0.31 | -0.79 | 2.3 | 5.0E-01 |
| *Lycopodium* | HF-HCl | -30.2 | 0.67 | 32 | 0.12 | 0.0 | 0.01 | 28.34 | 9.9E-01 | 0.362 | 2.58 | 2.7 | 9.1E-02 |
| *Pinus* | HF-HCl | -29.8 | 0.90 | 27 | 0.18 | -0.7 | -2.64 | 50.80 | 1.1E-02* | -0.38 | -1.95 | 3.3 | 1.4E-01 |
| *Zea* | HF-HCl | -14.7 | 1.03 | 31 | 0.22 | -1.4 | -4.75 | 63.92 | 1.2E-05*** | -0.17 | -0.44 | 2.29 | 7.0E-01 |
| *Betula* | Saponification | -26.1 | 0.84 | 44 | 0.14 | 0.0 | 0.07 | 11.83 | 9.4E-01 | 0.34 | 2.05 | 2.47 | 1.5E-01 |
| *Eucalyptus* | Saponification | -27.5 | 0.79 | 22 | 0.19 | 0.3 | 0.89 | 21.15 | 3.8E-01 | 0.02 | 0.12 | 3.46 | 9.1E-01 |
| *Lolium* | Saponification | -27.9 | 1.17 | 25 | 0.28 | 0.4 | 0.86 | 36.03 | 4.0E-01 | 0.55 | 1.88 | 3.40 | 1.5E-01 |
| *Lycopodium* | Saponification | -30.1 | 0.7 | 42 | 0.12 | 0.0 | -0.02 | 24.35 | 9.9E-01 | 0.34 | 2.43 | 2.49 | 1.1E-01 |
| *Pinus* | Saponification | -28.9 | 0.92 | 26 | 0.20 | 0.1 | 0.38 | 55.99 | 7.0E-01 | 0.03 | 0.14 | 3.30 | 8.9E-01 |
| *Zea* | Saponification | -13.1 | 1.8 | 25 | 0.49 | 0.1 | 0.16 | 89.15 | 8.8E-01 | 2.27 | 4.82 | 5.73 | 3.3E-03** |
| *Betula* | Untreated | -26.1 | 1.2 | 22 | 0.28 | | | | | 0.11 | 0.37 | 4.50 | 7.3E-01 |
| *Eucalyptus* | Untreated | -27.7 | 1.2 | 23 | 0.33 | | | | | -0.16 | -0.47 | 4.24 | 6.6E-01 |
| *Lolium* | Untreated | -28.2 | 1.27 | 20 | 0.31 | | | | | 0.12 | 0.35 | 3.62 | 7.5E-01 |
| *Lycopodium* | Untreated | -30.1 | 0.84 | 24 | 0.17 | | | | | 0.31 | 1.63 | 3.21 | 2.0E-01 |
| *Pinus* | Untreated | -29.0 | 0.80 | 23 | 0.17 | | | | | 0.14 | 0.71 | 2.98 | 5.3E-01 |
| *Zea* | Untreated | -12.4 | 0.8 | 23 | 0.18 | | | | | 0.27 | 0.96 | 2.44 | 4.2E-01 |







**Table 3.** LA-IRMS mean peak area and standard deviation (SD) for each species-treatment combination, along with the SD of the PE standard corresponding to the respective mean peak area (Van Roij et al., 2017). An F-test was performed to compare the sample SD with the PE standard SD. * = p<0.05, ** = p<0.01, *** = p<0.001.

| Species | Treatment | Mean peak area (Vs) | SD LA | SD PE | F-statistic | $df_1$ | $df_2$ | *p* value |
|---|---|---|---|---|---|---|---|---|
| *Betula* | Acetolysis | 1.01 | 1.14 | 0.68 | 2.80 | 18 | 40 | 6.6E-03** |
| *Eucalyptus* | Acetolysis | 2.47 | 0.96 | 0.63 | 2.33 | 22 | 24 | 4.7E-02* |
| *Lycopodium* | Acetolysis | 3.32 | 0.36 | 0.44 | 0.68 | 23 | 9 | 4.4E-01 |
| *Zea* | Acetolysis | 6.86 | 1.06 | 0.46 | 5.32 | 26 | 27 | 4.7E-05*** |
| *Pinus* | Acetolysis | 1.31 | 0.69 | 0.71 | 0.95 | 24 | 34 | 9.0E-01 |
| *Lolium* | Acetolysis | 2.88 | 0.74 | 0.35 | 4.45 | 41 | 16 | 2.2E-03** |
| *Betula* | HF-HCl | 1.19 | 0.47 | 0.68 | 0.48 | 17 | 40 | 1.1E-01 |
| *Eucalyptus* | HF-HCl | 1.02 | 1.17 | 0.68 | 2.99 | 23 | 40 | 2.4E-03** |
| *Lycopodium* | HF-HCl | 3.91 | 0.65 | 0.44 | 2.17 | 31 | 9 | 2.2E-01 |
| *Zea* | HF-HCl | 6.29 | 0.80 | 0.46 | 3.05 | 30 | 27 | 4.5E-03** |
| *Pinus* | HF-HCl | 1.75 | 0.91 | 0.55 | 2.71 | 26 | 19 | 2.9E-02* |
| *Lolium* | HF-HCl | 1.06 | 0.78 | 0.68 | 1.30 | 21 | 40 | 4.6E-01 |
| *Betula* | Saponification | 2.62 | 0.84 | 0.35 | 5.72 | 43 | 16 | 4.5E-04*** |
| *Eucalyptus* | Saponification | 5.76 | 0.76 | 0.46 | 2.73 | 21 | 23 | 2.1E-02* |
| *Lycopodium* | Saponification | 2.04 | 0.65 | 0.63 | 1.05 | 41 | 24 | 9.1E-01 |
| *Zea* | Saponification | 7.53 | 1.71 | 0.39 | 19.17 | 25 | 20 | 6.7E-09*** |
| *Pinus* | Saponification | 1.41 | 0.92 | 0.71 | 1.68 | 25 | 34 | 1.6E-01 |
| *Lolium* | Saponification | 2.47 | 1.13 | 0.63 | 3.24 | 24 | 24 | 5.6E-03** |
| *Betula* | Untreated | 0.73 | 1.15 | 1.05 | 1.21 | 21 | 20 | 6.8E-01 |
| *Eucalyptus* | Untreated | 0.76 | 1.17 | 1.05 | 1.24 | 22 | 20 | 6.3E-01 |
| *Lycopodium* | Untreated | 0.97 | 0.84 | 0.68 | 1.51 | 23 | 23 | 3.3E-01 |
| *Zea* | Untreated | 6.54 | 0.82 | 0.46 | 3.18 | 22 | 27 | 5.0E-03** |
| *Pinus* | Untreated | 1.94 | 0.80 | 0.52 | 2.37 | 22 | 21 | 5.3E-02 |
| *Lolium* | Untreated | 1.32 | 1.27 | 0.71 | 3.18 | 19 | 34 | 3.2E-03** |

## 4 Discussion

### 4.1 The overall chemical composition of (un)treated sporomorphs

*Untreated*

All the untreated sporomorphs are richest in lipids when examined by flash pyrolysis, mostly consisting of $C_{16}$ and $C_{18}$ fatty acids. Polysaccharide- and protein-derived products are the second most abundant compound group in the pollen of *Betula*, *Lolium* and *Zea*, while they are relatively small in *Eucalyptus* and *Pinus*, with only traces found in *Lycopodium* spores. Almost in opposite order are the relative abundances of aromatic pyrolysis products: richest in *Lycopodium* and *Pinus*, followed by *Betula* and *Eucalyptus* and very small abundances in the pollen of *Lolium* and *Zea* (Fig. 2).

Aromatics consist of various alkyl- and methoxyphenols, alkyl- and carbonylbenzenes, and vinyl containing compounds (see section 4.2.). Polysaccharides are dominated by levoglucosan (Fig. S3), a well-known pyrolysis product of cellulose





(Pouwels et al., 1987), which forms the main component of the intine of sporomorphs (Shim et al., 2022). Moreover, the
presence of xylose- and mannose-derived pyrolysis products suggests the sporomorphs potentially contain a large variety of
polysaccharides in addition to cellulose, considering this suite of monosaccharide building blocks. The N-bearing pyrolysis
products are all well-known pyrolysis products of proteins (Tsuge and Matsubara, 1985).

It is important to note that the Py-GC-MS method used in this study has a bias toward detecting apolar structures (i.e.,
the lipids) due to the apolar GC column used. Consequently, polar sugars are under-detected. Whether aromatics are under-
represented depends on the degree of polar groups (e.g., COOH, CO, OH) associated with the molecule. However, considering
pyrolysis is a well-established method for assessing the chemical composition of organic material, we consider this a robust
representation of sporomorph composition, especially when comparing to other studies using pyrolysis.

In sum, untreated sporomorphs are dominated by lipids (i.e., fatty acids) followed by polysaccharides and a mixture of
aromatics, with the exact composition varying among the studied species.


### *Saponification*

Upon saponification relative abundances of the extractable and esterified lipids decrease substantially compared to other
pyrolysis products (Fig. 2). This treatment not only eliminates alkenes, alkanes, alcohols and fatty acids but also removes some
aromatic acids, to varying extents, across all pollen types (Supplement Fig. S4). However, in *Lycopodium*, the aromatic acids
(i.e., hydrocaffeic and hydroferulic acid) appear to be relatively persistent (Supplement Fig. S4). The mixture of aliphatic and
aromatic compounds removed likely constitutes a significant component of pollenkitt, the sticky substance on pollen and
spores that plays a crucial role in *e.g.* the formation of pollen clumps and adherence to pollinators (Pacini & Hesse, 2005).

### *HF-HCl*

When the sporomorphs are treated with HF-HCl the relative contribution of polysaccharides and proteins becomes small to
very small for all HF-HCl treated sporomorphs (Fig. 2), supposedly due to acid-catalyzed hydrolysis of the glycosidic and
peptide bonds, respectively. This implies that the HF-HCl treatment effectively removes polysaccharides and proteins. The
relative amounts of the lipids detected are hardly affected, likely due to the very polar solvent which prevents solubilization of
the lipids, which prefer to stay in the vicinity of the hydrophobic sporomorphs rather than in the aqueous solution.


### *Acetolysis*

Due to the acidic conditions of acetolysis, polysaccharides and proteins are removed, resembling the HF-HCl treatment.
Moreover, extractable and hydrolyzed lipids are removed with methanol in the water removal step, leaving sporomorphs rich
in aromatic compounds (Fig. 2). These aromatic components detected after acetolysis therefore likely predominantly reflect
the pyrolysis products of sporopollenin.



## 4.2 The aromatic make-up of sporopollenin

Given that the aromatic components present after acetolysis primarily reflect the pyrolysis products of sporopollenin, a more detailed examination of the aromatic composition of sporopollenin is insightful. Nierop et al. (2019) showed that *Lycopodium* spores are rich in cinnamic acid derivatives, such as *p*-coumaric acid, caffeic acid and ferulic acid, and their respective saturated

analogues, phloretic acid, hydrocaffeic acid and hydroferulic acid. These components reflect typical sporopollenin-derived components (Jardine et al., 2017, 2021; Li et al., 2019; Lutzke et al., 2020).

Hydroferulic and hydrocaffeic acid are relatively polar structures, which do not have optimal interactions with the apolar GC column used. However, apparently *Lycopodium* produces these compounds in such abundances (Nierop et al., 2019) that they were still detected (Fig. S4).

Upon pyrolysis the unsaturated acids (i.e., *p*-coumaric acid, caffeic acid and ferulic acid) decarboxylate, or when ester-bound yield, 4-vinylphenol, 4-vinylcatechol and 4-vinylguaiacol, respectively, in contrast to the saturated acids (Moldoveanu, 1998). The presence of 4-vinylphenol and 4-vinylguaiacol in all sporomorphs studied here thus suggests their precursors *p*-coumaric acid and ferulic acid to be a consistent component of the sporopollenin polymer for all species. The relative contribution of 4-vinylphenol and 4-vinylguaiacol varied across sporomorph species/treatment combinations in such a way

that their precursors *p*-coumaric and ferulic acid must occur in hydrolysable and non-hydrolysable forms.

Based on Thermally assisted Hydrolysis and Methylation-GC-MS (TMH-GC-MS) analyses of spores from eight species of moss, clubmoss, horsetail and fern, as well as pollen from a single cycad species, Nierop et al. (2019) assumed that caffeic acid synthesis in sporomorphs might have been lost during Cycadophyta evolution but was retained in Magnoliophyta (angiosperms). However, 4-vinylcatechol, and thus its precursor caffeic acid, was found only in the clubmoss *Lycopodium*,

and was absent from the four angiosperms (*Betula*, *Lolium*, *Zea* and *Eucalyptus*), as well as from the gymnosperm *Pinus* (this study) and *Zamia floridana* (Nierop et al., 2019). These findings suggest that caffeic acid synthesis in sporopollenin is specific to spore-producing seedless embryophytes (mosses, clubmosses, liverworts, hornworts, horsetails and ferns) as opposed to pollen-producing seed plants.

Two other components, benzaldehyde and acetophenone, have been regularly identified in all kinds of samples upon

pyrolysis in relatively small abundances (Moldoveanu, 1998). In our spores and pollen these components are found in remarkably high abundances independent of treatment, and are most enriched in the pyrolysates after acetolysis, except for *Eucalyptus* (Fig. S3). Also in the extant megaspores of the lycophyte *Isoëtes killipii* (Boom et al., 2005) and megaspores and massulae of the waterfern *Azolla caroliniana* (Nierop et al, 2011) and in particular in the fossil spores of *Azolla* and *Salvinia* (Van Bergen et al., 1993) both pyrolysis products are prominently present. Furthermore, Boom et al. (2005) synthesized a

sporopollenin-like dehydrogenation polymer based on *p*-coumaric acid. Upon pyrolysis this dehydrogenation polymer yielded almost the same pyrolysis product as purified megaspores of *I. killipii* but lacked both benzaldehyde and acetophenone. This suggests that although *p*-coumaric acid is indeed an important constituent, additional aromatic components, associated with acetophenone and benzaldehyde as pyrolysates, must also be part of the sporopollenin structure (cf. Li et al., 2018).





Another series of components identified, guaiacol (2-methoxyphenol), 4-methyl- and 4-ethylguaiacol, together with the
aforementioned 4-vinylguaiacol (Supplement Fig. S4), are typical products found when lignin is present. However, the most
distinct pyrolysis products of intact lignin, the three isomers of 4-propenylguaiacol, were not detected (Saiz-Jimenez and De
Leeuw, 1984; Ralph and Hatfield, 1991). Hence, guaiacol, 4-methylguaiacol, and 4-ethylguaiacol likely originate from another
polyphenol source. The most probable contributors are ferulic acid-rich sporopollenin (*Lycopodium*) or tannin-rich species
such as *Betula* and *Eucalyptus* (Galetti and Reeves, 1992). These compounds are present in the other three species in
substantially lower amounts (Supplement Fig. S4).

Alkylbenzenes and alkylphenols can have multiple sources such as tannins and proteins (Tsuge and Matsubara, 1985,
Bracewell and Robertson, 1984; Galletti and Reeves, 1992). Since all N-bearing components (i.e., proteins) were absent after
HF-HCl and acetolysis treatments, the relative accumulation of alkylbenzenes and alkylphenols upon treatment implies a major
sporopollenin source of these pyrolysis products.

**4.3 The $\delta^{13}$C composition of (un)treated sporomorphs**

*Overall isotopic composition*

The $\delta^{13}$C value of the sporomorphs is essentially a weighted average of the $\delta^{13}$C of all the aforementioned chemical
components, broadly divided into polysaccharides/sugars, lipids, proteins and aromatic structures (cf. Hayes, 2001). Except
for *Eucalyptus* and *Lolium*, untreated sporomorph species differ significantly from each other in their mean $\delta^{13}$C values (Fig.
3), reflecting the different chemical compositions among the studied species, as well as differences in altitude, humidity,
photosynthetic pathways ($C_4$ [*Zea*] *vs.* $C_3$ [other species]) and other environmental growth conditions that influence carbon
isotope fractionation, from the $CO_2$ that is taken up to the sporomorph formed (Diefendorf et al., 2010; Farquhar et al., 1982;
O'Leary, 1981).

*Saponification*

The general $\delta^{13}$C increase in saponified sporomorphs compared to untreated samples aligns with the fact that lipids typically
have $\delta^{13}$C values 3 – 6‰ lower than bulk biomass (Abelson & Hoering, 1961; Hayes, 2001). This offset reflects the
fractionation along the metabolic pathways involved in lipid synthesis which uses acetyl co-enzyme A (acetyl-CoA) as the
carbon donor for the lipid skeleton (Hayes, 2001). Acetyl-CoA is produced through pyruvate dehydrogenase, which is known
to generate substantial isotopic fractionation (DeNiro & Epstein, 1977; Melzer & Schmidt, 1987). Removal of lipids thus
results in the $\delta^{13}$C value of the remaining material increasing. This isotopic effect of saponification contrasts with the findings
of Müller et al. (2021), who found a relative $\delta^{13}$C drop of pollen after KOH treatment. Evidently, the use of methanol and
DCM in addition to KOH during our saponification treatment resulted in a more effective extraction of lipids, causing the
observed $\delta^{13}$C increase. The LA-IRMS measurements of saponified samples show no significant rise in $\delta^{13}$C relative to
untreated pollen, which is likely due to the relatively small sample size ($n$ = ~30, as compared the EA-IRMS method, which



measures >1000 sporomorphs in a single measurement), causing the standard errors of the means to be too large to statistically detect a potential significant offset.

*HF-HCl*

Considering most of the hydrolysable compounds of the sporomorph consist of polysaccharides, which are generally $^{13}$C-enriched as compared to total biomass (Hayes, 2001; van Dongen et al., 2002), HF-HCl treatment logically results in lower sporomorph $\delta^{13}$C. Proteins generally have high $\delta^{13}$C compared that of total biomass (Hayes, 2001; Abelson and Hoering, 1961). However, the $\delta^{13}$C variation among individual amino acids is large (up to 25‰, Abelson and Hoering, 1961; Macko et al., 1987). This variation arises from the interplay between the sources of the carbon skeletons (e.g., the tricarboxylic acid cycle,

pyruvate, or phosphoglycerate) and the isotopic fractionation occurring at metabolic branch points during amino acid biosynthesis (see Hayes [2001] for an in-depth discussion on $^{13}$C fractionation during amino acid biosynthesis). Polypeptide $\delta^{13}$C can therefore theoretically be higher, lower or similar to total biomass $\delta^{13}$C values (Blair et al., 1985; Abelson and Hoering, 1961; Benner et al., 1987). Due to the relatively small amount of N-bearing compounds present in the sporomorphs, the isotopic effect of HF-HCl is rather dominated by the removal of polysaccharides, i.e., a depletion of the residual

sporomorph. The removal of polysaccharides clearly has a substantial effect on $\delta^{13}$C values (HF-HCl treatment in Fig. 3), even though the pyrolysis based chemical compositions of HF-HCl treated and untreated samples appear similar (Fig. 2). This is likely due to the aforementioned under-detection of polysaccharides with the Py-GC-MS method applied.

*Acetolysis*

Interestingly, for all species, acetolysis results in even more $^{13}$C-depleted sporomorphs than HF-HCl (Fig. 3), in line with Müller et al. (2021). This depletion suggests that the residual aromatic components in the sporomorph have a lower $\delta^{13}$C value than the extractable lipids. Genetic studies (Grienenberger et al., 2011; Zhu et al., 2021) in plants identified specific genes involved in exine formation. These genes encode reductase and synthase enzymes that catalyze the production of tetraketide α-pyrone, a cyclic compound, from medium- to long-chain fatty acids. Tetraketide α-pyrone is believed to serve as an important

precursor for sporopollenin suggesting that the precursor of sporopollenin is a fatty acid (Grienenberger et al., 2011; Kim & Douglas, 2013; Zhu et al., 2021). Like fatty acids and other lipids, the biosynthesis of tetraketide α-pyrones relies metabolically on acetyl-CoA, which, as described above, generates substantial $^{13}$C fractionation through the pyruvate dehydrogenase reaction involved in its production (DeNiro & Epstein, 1977; Melzer & Schmidt, 1987). Therefore, the relatively low $\delta^{13}$C values of the aromatic compounds as compared to lipids, can be explained by this additional acetyl-CoA induced fractionation.

**4.4 EA-IRMS versus LA-IRMS**

For the EA-IRMS measurements, some natural and analytical variability in $\delta^{13}$C values is evident from the spread in the untreated samples. Particularly for *Zea* and *Lolium*, which have a substantially larger spread than the standard deviation of the GQ standard (0.07‰), it is clear the untreated pollen were not entirely homogeneous before measuring, resulting in some





natural variability contributing to the observed variability in the EA-IRMS measurements. However, the even larger spread

recorded for acetolyzed pollen (e.g., *Pinus* and *Zea*) is likely due to non-uniform treatment of the material when, for example, the duration of contact between sporomorphs and the acetolysis solution differs for individual sporomorphs, affecting both the composition and amount of removed substances (Hemsley et al., 1993). Except for saponified *Zea* the mean LA- and EA-IRMS $\delta^{13}C$ measurements are not significantly different.

For saponified *Zea*, the significant difference between the LA- and EA-IRMS means suggests something is measured

differently among the two methods. Light microscopy photos show the coloration of the saponified pollen to be more variable than untreated pollen (Supplement Fig. S6), which is possibly represented by the skew to lower $\delta^{13}C$ values witnessed for the distribution of saponified *Zea*. However, the reason behind this EA/LA mismatch remains elusive. An additional PLA correction does not significantly change this result (Supplement Fig. S9).

Overall, LA-IRMS is the preferred method to study mean $\delta^{13}C$ of sporomorph populations due to more direct targeting

of individual or very few specimens, avoiding the isotopic incorporation of non-sporomorph material (e.g., anther fragments) also present in samples. Furthermore, the advantage of LA-IRMS over EA-IRMS is that it enables the analysis of inter-specimen variability and community structure. This is exemplified by the – larger than analytical error – spread of untreated *Zea* and *Lolium* in this study, representing intraspecific differences in carbon isotope ecology, and further highlighted by Frieling et al. (2023), Sluijs et al. (2018) and Van Roij et al. (2017).

**4.5  Implications for fossil sporomorphs**

The sporopollenin-rich sporomorph left behind after acetolysis can be thought of as being similar to diagenetically altered sporomorphs from the geological record: most labile and hydrolysable components are removed, and only the most durable material remains. However, fossil sporomorphs have been found to not only be rich in aromatic compounds, but also in aliphatic structures, depending on the stage of maturation (Van Bergen et al., 1993; Versteegh et al., 2007; Yule et al., 2000).

The *post-mortem* aliphatic groups found in fossil sporomorphs possibly originate from the abundant fatty acids present on the exine during early diagenesis (De Leeuw et al., 2006), whereas in higher mature stages sporomorph aromatization may occur, with a loss of aliphatic hydrocarbons (Yule et al., 2000). This implies sporomorph diagenesis may result in stable carbon isotopic fractionation to values between as observed for HF-HCl treated (i.e., dominated by lipids and aromatics) and acetolyzed (i.e., dominated by aromatics) sporomorphs, depending on the stage of maturation (Yule et al., 2000). Further

investigations focusing on carbon isotope analysis of HF-HCl treated spores that underwent hydrous pyrolysis to mimic maturation (cf. Watson et al., 2012), could shed light on this.

Acetolysis may seem ideal for isolating sporopollenin from modern sporomorphs, allowing appropriate comparison with fossil specimens. However, it should be emphasized that the acetolysis procedure seems to affect the material non-uniformly (Hemsley et al., 1993), which increases the variation between individual sporomorphs and hence the error in $\delta^{13}C$

measurements (Fig. 4). Therefore, for modern sporomorphs, a combination of HF-HCl (removing sugars and proteins) followed by a lipid removal step (e.g., washing with methanol) could be a good alternative to acetolysis for isolating





sporopollenin. However, considering the removal of lipids only minimally affects $\delta^{13}$C values, as witnessed for saponified compared to untreated sporomorphs (Fig. 3), we suggest that the HF-HCl procedure alone is an appropriate treatment when comparing $\delta^{13}$C values among modern and fossil sporomorphs.

## 5 Conclusions

The chemical composition of modern sporomorphs as determined by flash pyrolysis is dominated by lipids (i.e., mostly $C_{16}$ and $C_{18}$ fatty acids), followed by aromatics and polysaccharides, the order of which depends on plant species, with minimal amounts of proteins. After removal of extractable and esterified lipids and hydrolysable polysaccharides and proteins, the sporomorph is dominated by aromatic compounds, which are thought to represent the components of the resistant sporopollenin structure. In addition to present literature, our study shows the composition of this sporopollenin is variable among species. However, *p*-coumaric and ferulic acid are consistent building blocks, whereas the carbonylbenzenes benzaldehyde and acetophenone seem to be unique markers of sporopollenin upon pyrolysis.

Overall, the LA- and EA-IRMS methods agree, meaning the average $\delta^{13}$C of individually measured sporomorphs is in the line with the bulk value. However, LA-IRMS is the preferred method for $\delta^{13}$C measurements due to more accurate targeting of the sporomorphs, the possibility to study inter-specimen variability and community structure, and the low amounts of material needed to get an accurate measurement. To take linear mass bias into account when correcting the data, we do advise to use multiple standards across a range of $\delta^{13}$C values that ideally bracket the range of the data (e.g., PE and PLA).

Both LA- and EA-IRMS illustrate that saponification slightly increases $\delta^{13}$C values by removal of surface-bound lipids, HF-HCl decreases $\delta^{13}$C values by removal of polysaccharides and proteins, and acetolysis causes an even more pronounced decrease in $\delta^{13}$C values, by concentrating the relatively $^{13}$C depleted sporopollenin. A combination of HF-HCl treatment and a lipid removal step appears to be the most effective method for investigating the $\delta^{13}$C value of sporopollenin specifically, as acetolysis affects the material in an overly non-homogeneous manner. Fossil sporomorphs likely have a $\delta^{13}$C value somewhere between HF-HCl treated and acetolyzed samples of fresh sporomorphs, depending on the stage of maturation, which influences the relative amount of aliphatic lipids as compared to aromatics.

*Data availability.* All newly generated data will be available via a permanent online repository (Zenodo https://doi.org/10.5281/zenodo.15174422, Bats, 2025) upon publication.

*Author contributions.* AS and KGJN designed the study. LvR, ASL, JF, KGJN and YFB processed samples and generated data. YFB, KGJN and AS analyzed data. YFB wrote the original draft. AS, KGJN, JF and GJR reviewed and edited the paper. AS and GJR received funding for this study.



*Competing interests.* The communicating author has declared that none of the authors has any competing interests.

*Acknowledgements.* We thank Arnold van Dijk, Desmond Eefting, Natasja Welters and Giovanni Dammers (Utrecht
University) for technical and analytical assistance, and Francien Peterse for valuable discussions.

*Financial support.* We thank the Ammodo Foundation for funding unfettered research by laureate Appy Sluijs. Appy Sluijs
acknowledges funding from European Research Council ERC consolidator grant 771497 (SPANC). This work was supported
by EMBRACER (Summit Grant SUMMIT.1.034), financed by the Netherlands Organization for Scientific Research (NWO).
This work was carried out under the program of the Netherlands Earth System Science Centre (NESSC); has been financially
supported by the Ministry of Education, Culture, and Science (OCW) through Gravitation (grant no. 024.002.001); and has
received funding from the European Union's Horizon 2020 research and innovation program under the Marie Skłodowska-
Curie Actions (grant no. 847504).

*Review statement.*

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
