# Peer review of "Impact of chemical treatments on the molecular and stable carbon isotopic composition of sporomorphs"

_EGUsphere, 2025_

## Author Response (AR1)

Dear Editor,

We thank you and the reviewers for the positive evaluation of our manuscript. Below we address each review comment with a description of its implementation in our revised manuscript. We also provide a summary list of all changes made in our revised manuscript.

Sincerely, also on behalf of my co-authors,

Yannick Bats, MSc.

Comments by referee 1:

*This is a clear very well written paper looking at an interesting and important aspect of paleo science.*

*I offer a couple of minor comments,:*

*The discussion from line 400 about the differences between sporopollenin from seed plant and non-seed plants seems to be resented with too much certainty given the very limit amount of taxa currently evaluated. I would recommend this statement be revised o reflect uncertainty.*

Implementation:

We added the following sentence: "However, considering the limited amount of taxa studied here, more research is needed to confirm this."

*I think the section 4.5 should be expanded and a more detailed comparison made to data from the experiments carried out in this study and to experiments that have sort to tease out changes in chemistry by simulating thermal maturation and maybe this could be extended to look at fossils. This would give a more rounded approach to the discussion and make the paper more useful to the broader bio-geoscience community.*

Implementation:

We expanded section 4.5 with a comparison with experiments that simulated thermal maturation, specifically Watson et al. (2012), who did hydrous pyrolysis (to 350 °C) on spores, and Yule et al. (2000), who heated spores to 300 °C. We discuss the stepwise change in chemical composition of sporopollenin observed through thermal maturation, influencing the aliphatic to aromatic proportion of sporopollenin. We relate these changes to carbon isotopic changes as observed for our acetolyzed and HF-HCl treated sporomorphs.

*With regard to figures I would encourage the authors to reconsider their colour palette choices and avoid red green color combinations given the prevalence of red/ green color vision deficiency.*
Implementation:
We changed the figures to a colorblind friendly color scheme based on blue, purple and green tints.
*I wonder if figure two might be better plotted as a difference from untreated for each taxa.*
Implementation:
We have experimented with replotting Figure 2 with values as a difference from untreated, but all of our efforts resulted in less clear figures. We therefore respectfully decided to retain the current version, but with colorblind friendly colors.

Comments by referee 2:

*This is an interesting and well-written paper, focusing on sporopollenin δ13C measurements across a series of taxa, in relation to different laboratory processing procedures and analytical approaches (EA-IRMS vs LA-IRMS).*

*I think the paper is broadly publishable as is it. The only part I am less convinced by is section 4.5 'Implications for fossil sporomorphs'. The authors treat acetolysis as if it efficiently isolates the sporopollenin wall with no additional effects, but previous studies have shown that acetolysis also alters the sporopollenin: Lutzke et al. (2020) and Wang et al. (2023) showed that acetolysis reduces the phenolic content of sporopollenin, and Amundsen et al. (1997) and Loader and Hemming (2000) showed a considerable decrease in δ13C values with acetolysis, including (in Loader and Hemming) in relation to a different sporopollenin isolation approach (sulphuric acid - so it's not just that the δ13C value goes down because the sporopollenin is being isolated, the acetolysis is actually doing something else as well). Dominguez et al. (1998) and Jardine et al. (2015, 2017, 2021, 2023) have also used FTIR to show that new peaks appear with acetolysis that are not produced by other processing/isolation approaches. So acetolysis is useful for removing labile compounds and leaving the sporomorph exine behind for morphological analyses, but from a chemical point of view it is does change things, and I would be wary of saying that it produces something similar to diagenetically altered sporomorphs from the geological record (the authors also need to be careful here - do they mean relatively recently buried sporomorphs, where we might expect the labile compounds to have degraded but the sporopollenin to be more or less unchanged, or geological samples where the sporopollenin has repolymerised?).*

Implementation:

We expanded section 4.5 with a concise discussion on the acetolysis method. We discuss the findings of other studies (Dominguez et al., 1998; Jardine et al., 2017, 2023; Wang et al, 2023) that acetolysis alters sporopollenin by addition and removal of C-bearing compounds. We discuss the fact that this will most likely affect the $\delta^{13}C$ value of the residual sporopollenin, where we refer to our own results (which show extreme $^{13}C$-depletion associated with acetolysis), and other work where acetolysis was compared to (other) sporopollenin isolating techniques (Amundson et al., 1997; Loader and Hemming, 2000). We emphasize that we did not use acetolysis to isolate sporopollenin, and we advise against it for this purpose. Instead, we advise to use other non C-bearing techniques (Amundson et al., 1997; Loader and Hemming, 2000; Li et al., 2019; Lutzke et al., 2020), when the aim is to isolate sporopollenin.

*In terms of recommendations for comparing modern and fossil sporomorphs, the authors also need to keep in mind that other approaches for isolating sporopollenin have been suggested. Li et al. (2019) and Lutzke et al. (2020) used a combination of enzymes and solvents, for example, and Loader and Hemming (2000) used sulphuric acid. Jardine et al. (2023) carried out a comparison of these and other methods from the perspective of FTIR and chemotaxonomy. I don't suggest that the authors add these approaches to this study, but some comments in the discussion to point out that other methods are available and could be compared in future research would be a useful addition.*

*There are three papers that I have mentioned here but are not cited in the paper, which I think the authors would do well to read and incorporate into the text:*

*Domínguez, E., J. A. Mercado, M. A. Quesada, and A. Heredia. 1998. Isolation of intact pollen exine using anhydrous hydrogen fluoride. Grana 37(2):93-96.*

*Jardine, P. E., M. S. Kent, W. T. Fraser, K. H. Knorr, and B. H. Lomax. 2023. Uncovering a phylogenetic signal in plant biopolymer chemistry: a comparison of sporopollenin isolation approaches for use in palynological research. Palaeontology 66(6): e12683.*

*Wang, T., Bell, B.A., Fletcher, W.J., Ryan, P.A., and Wogelius, R.A. 2023. Influence of common palynological extraction treatments on ultraviolet absorbing compounds (UACs) in sub-fossil pollen and spores observed in FTIR spectra. Frontiers in Ecology and Evolution 11:1096099.*

Implementation:

We expanded section 4.5 with a comparison with experiments that simulated thermal maturation, specifically Watson et al. (2012), who did hydrous pyrolysis (to 350 °C) on spores, and Yule et al. (2000), who heated spores to 300 °C. We discuss the stepwise change in chemical composition of sporopollenin observed through thermal maturation, influencing the aliphatic to aromatic proportion of sporopollenin. We relate these changes to carbon isotopic changes as observed for our acetolyzed and HF-HCl treated sporomorphs.

*And finally, one minor comment:*
*Lines 42 - 43: exine isn't a common term for sporopollenin - the exine is the outer wall of pollen and spores, and is composed of sporopollenin. I suggest rephrasing this sentence.*

Implementation:
Changed to: "Organic microfossils are comprised of organic macromolecular structures such as sporopollenin (the main component of exine – the outer wall of pollen and spores), residual labile compounds (e.g., polysaccharides, proteins) and diagenetically produced 'geopolymers' that become enriched over time and are only susceptible to strong oxidation (De Leeuw et al., 2006; Quilichini et al., 2015)."

List of all relevant changes made in the manuscript:

1. Line 42, sentence rewritten to: "Organic microfossils are comprised of organic macromolecular structures such as sporopollenin (the main component of exine – the outer wall of pollen and spores), residual labile compounds (e.g., polysaccharides, proteins) and diagenetically produced 'geopolymers' that become enriched over time and are only susceptible to strong oxidation (De Leeuw et al., 2006; Quilichini et al., 2015)."
2. Line 414, sentence added to clarify uncertainty: "However, considering the limited amount of taxa studied here, more research is needed to confirm this."
3. Figure color scheme changed to colorblind friendly.
4. Section 4.5. modified to include a better comparison with other studies that conducted artificial maturation experiments, and a better discussion of the acetolysis method, also in comparison with (other) sporopollenin isolating techniques. More relevant literature is included for comparison.